# Integration experiences of internationally educated diagnostic radiographers working in the UK

**Elaine Wilkinson** [1]*, **Edozie Iweka** [2], **Beverley Snaith** [1,3], **David Omiyi** [1]

1 University of Bradford, Bradford, United Kingdom, 2 University Hospitals of Derby and Burton NHS Foundation Trust, Derby, United Kingdom, 3 Mid Yorkshire Teaching NHS Trust, Wakefield, United Kingdom

* e.chaplin@bradford.ac.uk

## Abstract

### Background

The current diagnostic radiography workforce issues in the UK have led to active recruitment strategies to employ radiographers from overseas. Research into the experience of nurses and other healthcare professions migrating to the UK to work in the NHS report challenges to that transition, however there is a paucity in the research into the experiences of internationally trained diagnostic radiographers.

### Method

This study aimed to explore the experiences of internationally educated diagnostic radiographers (IEDR) working in the UK and their intentions to remain. A UK-wide electronic survey was distributed through research and professional networks and social media in October 2023. Eligibility criteria were diagnostic radiographers, internationally educated, and currently working in the UK. 226 responses were received.

### Findings

There were positive responses related to the workplace support received (positive correlation r > .60 with statistical significance p < 0.05). However, 57.7% did not know what to expect when transitioning to work, and 56.0% experienced differences in societal culture that affected their adaptation to life and work in the UK. 30.2% had experienced workplace bullying and 34.2% felt they did not fit in culturally with the team. Family support, good workplace relationships and appropriate work mentoring were the top factors respondents identified as supporting integration into the UK. Over 40% intended to leave the UK or were uncertain if they would stay.

**Data availability statement:** Data cannot be shared publicly. The statement in the participanttient information sheet and in ethics application approved stated data access would be restricted to the researchers. Some data are available from the Ethics Committee (contact via ethics@bradford.ac.uk) for researchers who meet the criteria for access to confidential data.

**Funding:** The author(s) received no specific funding for this work.

**Competing interests:** The authors have declared that no competing interests exist.

## Conclusion

The findings indicate that cultural and societal acclimatisation is challenging, and the enormity of relocating to another country requires holistic support for individuals. More can be done to prepare and support the transition of internationally trained diagnostic radiographers into UK life and practice and improve retention.

## Introduction

The migration of healthcare professionals into the UK has occurred over many decades. Strategies using international recruitment have featured in workforce planning, alongside training and development, to ensure sufficient capacity to meet service delivery needs [1,2]. The initial Code of Practice for NHS Employers [3] was set out to ensure ethical recruitment practices. In addition, the World Health Organization has sought to safeguard countries where healthcare delivery would be disproportionately impacted from active recruitment by other governments [1,4]. Despite such codes of practice, migration of healthcare workers from countries where coordinated NHS recruitment is not sanctioned still occurs through individuals' own volition. Indeed, this pattern of migration of healthcare workers from low and middle to high income countries is seen globally [5,6]. The loss of skilled resources from a country, dubbed brain drain, is a phenomenon resulting from a number of push and pull factors including income, career opportunities, education and development, hospital infrastructure and management, working and living conditions, family and political issues [7,8].

It is estimated that the international workforce accounts for every 1 in 5 employees in the NHS [9]. In Saudia Arabia almost three quarters of doctors and nurses trained abroad, whilst in the USA this is 25% and 6–15% respectively [10]. Nurses and doctors, as larger professional groups, are often the target of recruitment campaigns, although more recently there has been an increase in allied health professionals. Indeed, radiographers are listed as an eligible healthcare profession on the UK government's skilled worker visa list [11]. As such, NHS England has supported NHS trusts with campaigns of international recruitment of diagnostic radiographers as another means to fill the workforce gap [12]. England has experienced a prolonged average vacancy rate of 11%, higher in some regions, for diagnostic radiographers [13,14]. In combination with the rising demand for diagnostic imaging services, this poses challenges in appropriate and timely service delivery [15,16]. Whilst the number of trainees has increased through NHS England workforce plans and departments supporting staff through apprenticeships, international recruitment remains a key part of workforce planning [1]. The NHS international recruitment toolkit [17] aims to support those leading and implementing international recruitment through the process from planning to beyond induction. The recruitment process can be challenging with obstacles around recognition of qualifications, visas, and registration with UK professional bodies [18–20]. Once in the UK, healthcare professionals are reported

to experience cultural challenges, difficulties with navigating UK systems, and barriers to transitioning into roles within the UK healthcare system [21–23].

Much of the existing research around the experiences of internationally educated healthcare professionals moving to the UK centers around the nursing and medical profession [18–20]. However there is a paucity in the research regarding the experiences of internationally educated diagnostic radiographers. Given the current UK climate of the radiography workforce deficit and the inclusion of international recruitment in the workforce planning strategy, alongside non-uniformity of training and scope of practice across the world, this study aims to explore the experiences of internationally educated diagnostic radiographers (IEDR) working in the UK and their intentions to remain. This will provide important findings for managers and policy makers given the effort to recruit and retain radiographers from oversees to stabilise workforce numbers and service delivery. Determining the backgrounds, drivers for migration, and radiography roles pre- and post-migration of IEDR was also part of the study aims, the findings of which have been reported in an earlier paper [24].

## Method

Ethical approval was obtained from the Research Ethics Panel at the University of Bradford on 15th September 2023 by the authors affiliated with that institution prior to study commencement (EC27952). Informed by a previous scoping review ([23], an online survey was developed with questions designed to ascertain the experiences of IEDR currently working in the UK and their intentions to remain, through multiple choice, Likert scale, ranking and free text responses (S1 Table). Additional demographic questions were included, those related to protected characteristics were optional, and other questions were not mandatory to encourage engagement. The findings of survey questions related to education, motivations for migration, and roles pre- and post-migration have been reported previously and therefore will not be detailed in this paper [24]. A participant information sheet was included introducing the survey purpose, why participants had been invited to take part, what participation involved, how data would be stored, access restricted, and reported, and contact details of the research team for further information or withdrawal from the study. Confirmation of reading the participant information sheet and written consent was obtained within the survey. The survey was piloted by five IEDRs and their feedback on the survey tool led to minor amendments to question wording to improve clarity and additionally changes to survey routing based on individual answers.

The survey was distributed via email to professional networks and via social media platforms including LinkedIn and X (formerly Twitter): the survey was open between 3rd October 2023–30th November 2023. A sample size calculation of 385 was based on an expected population of 4,186 [25] and a 5% margin of error. Data was downloaded from the survey platform into Excel (Microsoft Inc., US). Descriptive statistics using frequency and percentage distributions were used to summarize responses. Likert responses 'strongly agree' and 'agree' were amalgamated, as were 'strongly disagree' and 'disagree' to determine the overall positive, negative and neutral responses to the statements. Chi-square statistics were calculated for differences between categorical variables, Kruskal Wallis test for differences between ordinal and categorical variables and Spearman correlation for establishing the relationship between ordinal variables. Missing data was considered on a case by case basis. Both descriptive and inferential statistics were conducted using Statistical Package for Social Sciences (SPSS v26.0, IBM, US) with the level of significance set at 0.05. The open-ended questions were reviewed using thematic analysis. Responses were categorised into key themes to identify common perspectives and insights. Exemplar quotes have been included to illustrate the findings.

## Results

Two hundred and twenty-six participants met the eligibility criteria, and all completed the consent questions agreeing to take part and for the use of anonymised response data. An additional 5 responses were excluded due to incomplete survey responses (n = 3) and not positively responding to the consent to take part questions (n = 2). Africa was the largest originating continent (n = 108/226; 47.8%), and together with Asia (n = 70/226; 31.0%), making up nearly 80% of

participants (see Table 1). Black African was the largest ethnic group (n = 100/226; 44.2%). Over half of respondents were aged between 25–35 years old (n = 123/226; 54.4%). Nearly half of IEDR were married before arrival in the UK (n = 109/226; 48.2%) and this had risen to 62.4% at the time of the survey for those in the UK.

**Table 1. Demographics of participants (n = 226).**

| Demography | | Number (%) | |
|---|---|---|---|
| Age | 18-25 | 8 (3.5) | |
| | 26-35 | 123 (54.4) | |
| | 36-45 | 72 (31.9) | |
| | 46-55 | 17 (7.5) | |
| | 56-65 | 6 (2.7) | |
| Gender | Female | 101 (44.7) | |
| | Male | 124 (54.9) | |
| | Non-binary | – | |
| | Other | – | |
| | Prefer not to say | 1 (0.4) | |
| Marital status | | *Before arrival* | *At time of survey* |
| | Single | 99 (43.8) | 62 (27.4) |
| | Married | 109 (48.2) | 141 (62.4) |
| | Partnership/civil union | 10 (4.4) | 9 (4.0) |
| | Separated | 2 (0.9) | 3 (1.3) |
| | Divorced | – | 3 (1.3) |
| | Widowed | 1 (0.4) | 1 (0.4) |
| | Other | 1 (0.4) | 1 (0.4) |
| | Prefer not to say | 4 (1.8) | 6 (2.7) |
| Continent of Origin | Africa | 108 (47.8) | |
| | Asia | 70 (31.0) | |
| | Australasia | 7 (3.1) | |
| | Europe | 31 (13.7) | |
| | North America | 6 (2.7) | |
| | South America | 1 (0.4) | |
| | Blank | 3 (1.3) | |
| Ethnicity | Asian -Chinese | 2 (0.9) | |
| | Asian- Indian | 34 (15.0) | |
| | Asian – Pakistani | 1 (0.4) | |
| | White and Asian | 3 (1.3) | |
| | Any other Asian background | 34 (15.0) | |
| | Black -African | 100 (44.2) | |
| | Black –Caribbean | 2 (0.9) | |
| | Black -British | 2 (0.9) | |
| | White and Black African | 6 (2.7) | |
| | Any other mixed or multiple ethnic background | 3 (1.3) | |
| | Other | 36 (15.9) | |
| | Prefer not to say | 3 (1.3) | |

## Survey responses

Not all participants responded to every survey question, thus the number of respondents is indicated in the data. All the responses from participants indicating the support received on transition to work in the UK (Fig 1) showed strong positive correlations ($r > .60$) with statistical significance ($p < 0.05$). The only inconsistency in this domain was related to participants belonging to a social network that assisted in their adjustment to life and work in the UK, with 28.1% (n = 63/224) disagreeing with this. Participants had the highest agreement to having cordial work relationships with colleagues and managers (n = 199/225; 88.4%).

With regards to the work transition experiences of respondents (Fig 2), the majority of IEDR felt that their experience and skills were valued in the UK (n = 166/225; 73.8%) and were confident that they practiced at the required competency when started on their new job (n = 180/224; 80.4%). However, there were also perceived differences between clinical practice in their home country and the UK (n = 165/224; 73.7%). Similarly, a high number of respondents felt nervous adjusting to their UK work environment in their first few weeks (n = 175/224; 78.1%). The relationship between these two demonstrated a weak positive correlation, although statistically significant ($r = .14$, $p < 0.05$). There was also an admission by most respondents to struggling with the acronyms used in the UK clinical practice (n = 98/224; 43.8%).

Participants were asked to state what their intentions to stay in the UK were prior to arrival, and at the time of the survey. Those intending to stay in the UK long term increased from 54.9% (n = 124/226) to 58.8% (n = 133/226). Only 29/226 (12.8%) IEDR indicated they originally intended to stay in the UK for a short time before returning home, and 24/226 (10.6%) reported their current intention was to return to their home country. 31.0% (n = 70/226) were not sure what their intentions were prior to UK arrival, which reduced to 26.8% (n = 61/226).

The 3/226 (1.3%) radiographers who described their *original* intention to stay as 'other' gave additional explanation.

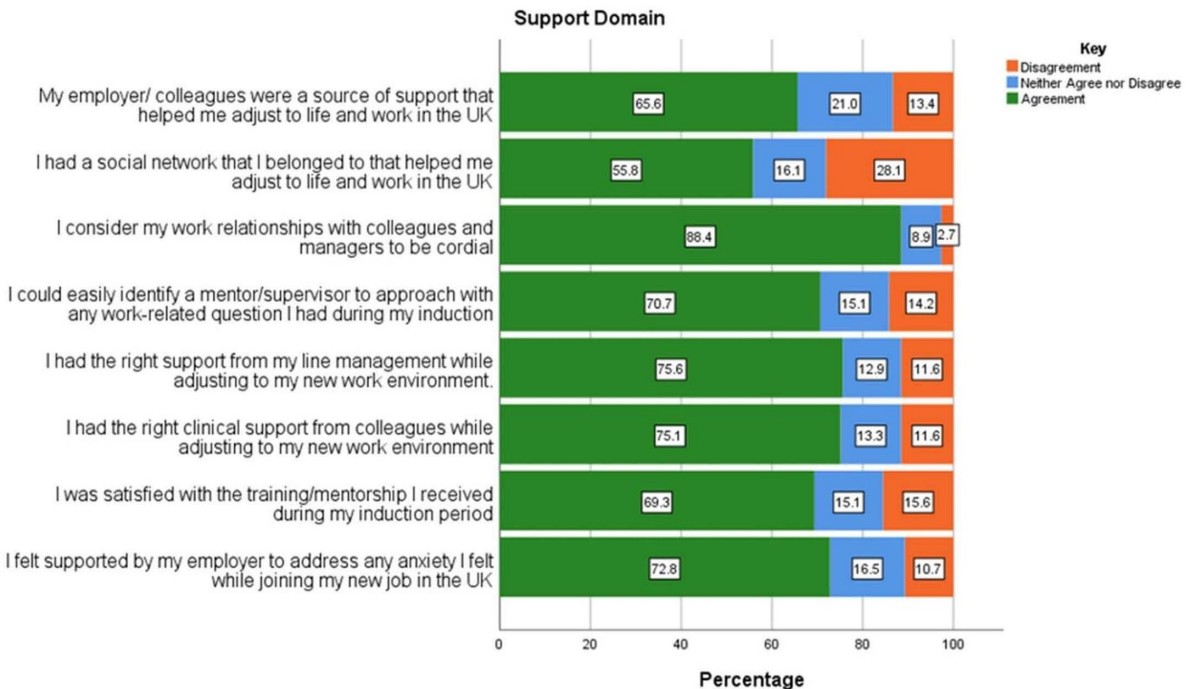

**Fig 1. Responses related to work placed support received by internationally educated diagnostic radiographers working in the UK.**

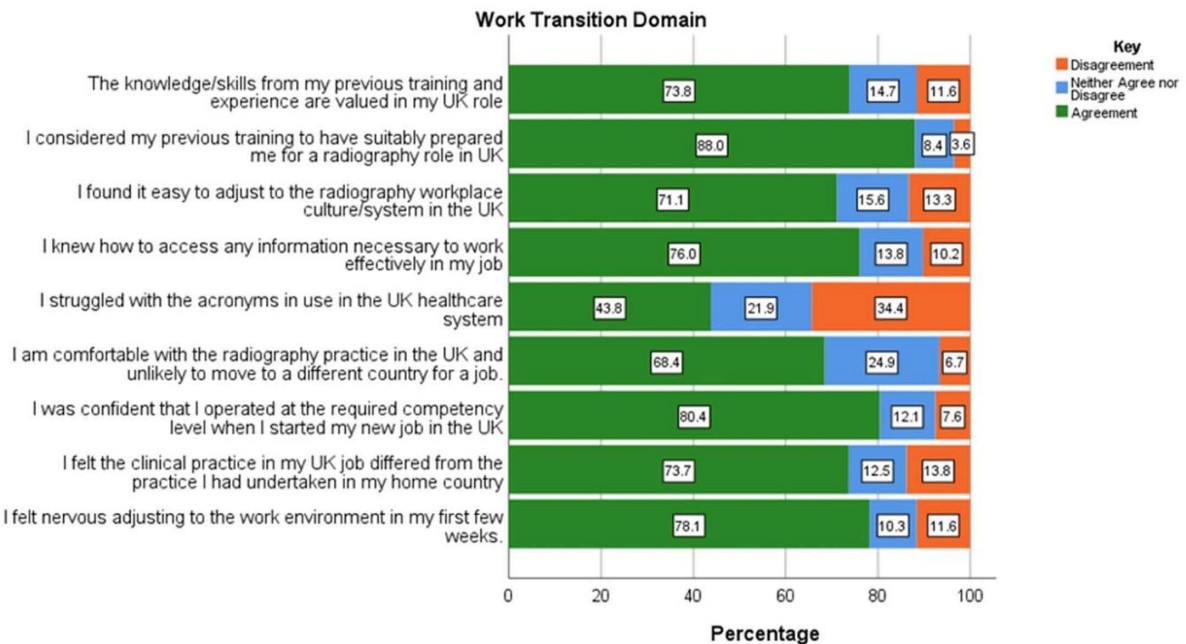

**Fig 2. Responses related to work transition of internationally educated diagnostic radiographers working in the UK.**

*'Initial plan was to stay for 5y [years]- it will be 10y next March'.* (ID195)

*'Test the waters first if all goes well, will stay for good'.* (ID177)

This rose to 8/226 (3.5%) when IEDR selected their current intentions to stay, with the free text responses including resigning from current role due to lack of respect for diversity by the employer (n = 1), returning home due to having a young child (n = 1), and the desire to relocating to another country (n = 6).

*"Relocate to another country eventually'* (ID115)

*'Move to North America'* (ID96)

Respondents personal transition experiences were the least positive compared with the other assessed domains (Fig 3): 30.2% (n = 68/225) experienced bullying because of their background as an international recruit, while 34.2% (n = 77/225) felt unable to "fit in" culturally within their team. A significant number of respondents did not know what to expect during the transition period (n = 130/225; 57.7%), with almost similar numbers experiencing differences in their new societal culture that affected their adaptation to life and work (n = 126/225; 56.0%). IEDR who identified as not having a social network more frequently stated that they were likely to stay in the UK on a short-term basis and return to home country (n = 10/24; 41.7%) than stay long term (n = 32/132;24.2%), however this was not statistically significant (p = 0.73). Furthermore, those who felt unable to "fit-in" culturally within the team and who felt Isolated after starting on their new role made up 70.8% (p < 0.05) and 75.0% (p = 0.73) respectively of those who see their stay in the UK as a short one before returning to their home countries. Experience of bullying due to participants' identification as an IEDR received the least agreement at 30.2% (n = 69/225), however a sub analysis identified that respondents from Africa (38.0%; n = 41/108), followed by Asia (24.6%; n = 17/69) are more likely to identify with this compared to other regions (p < 0.05).

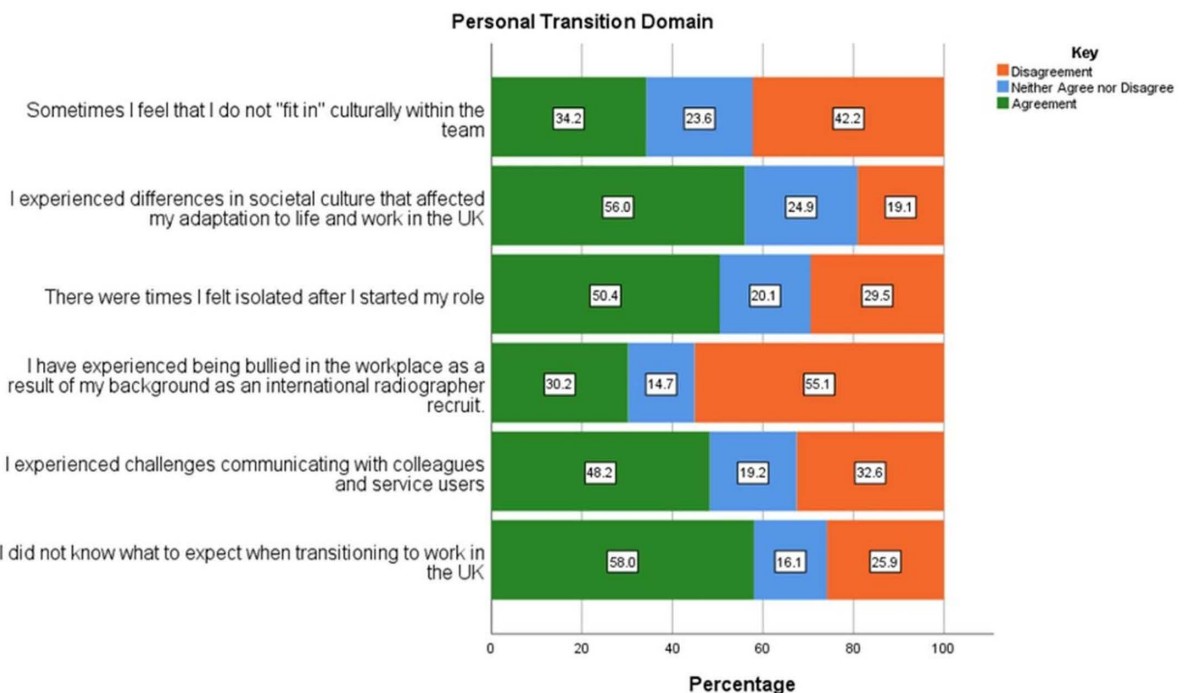

**Fig 3. Responses related to personal transition of internationally educated diagnostic radiographers working in the UK.**

There was minor gender differences noted in participants' response to integration questions: more male respondents (n = 97/124; 78.2%) found it easy adjusting to the radiography workplace culture and system in the UK compared to female respondents (n = 62/100; 62.0%). The difference noted between both genders on this question showed statistical significance ($H(2) = 7.02$; $P < 0.05$). Marital status was also found to be influential, for example, participants who were single on entry into the UK were more likely to have a social network that helped them adjust to life in the UK compared to those who were married (n = 63/97; 64.9% VS n = 54/109; 49.6%); less likely to feel Isolated on starting their new role (n = 44/98; 44.9% VS n = 63/108; 58.3%); less likely to feel bullied in the workplace (n = 21/98; 21.4% VS n = 40/109; 36.7%) and less likely to experience differences in cultures affecting adaptation to life and work in the UK (n = 50/98; 51.0% VS n = 68/109; 62.4%). However, statistical significance was only evident when marital status is correlated with feeling isolated after starting a new role ($H(2) = 6.6$; $P < 0.05$) and feeling bullied in the workplace ($H(2) = 7.3$; $P < 0.05$).

Respondents were asked to rank the top 5 factors which helped them integrate into the UK from a list of 16 options (Table 2). Seventeen respondents chose to rank less than 5 factors, and two respondents did not answer this question. A further 9 responses were disregarded due to missing numbers in the sequence of 5 meaning the intention of the responses could not be ascertained, leaving 215 respondents.

Twenty free text responses indicated additional factors which did or would have supported IEDR integration in the UK related. When analysed a number of themes were identified, and the frequency of these in the text comments is indicated, which included support from people or affiliations (n = 3), job related learning (n = 6), financial help (n = 3), processes at work (n = 2), and individual priorities or characteristics (n = 3).

*'Personal study time, connecting with other recently internationally recruited radiographers for learning support'* (ID112)

*'Community support from people with similar background'* (ID120)

**Table 2. Ranking of factors supporting integration into the UK (n = 215).**

| Factors | Number of responses by rank number | | | | | Total |
|---|---|---|---|---|---|---|
| | 1 | 2 | 3 | 4 | 5 | |
| Family support | 59 | 24 | 24 | 4 | 16 | 117 |
| Good workplace relationship | 14 | 24 | 27 | 24 | 14 | 103 |
| Appropriate work mentoring | 25 | 16 | 23 | 14 | 15 | 93 |
| Motivation to work in the UK, e.g., salary | 20 | 19 | 21 | 21 | 11 | 92 |
| Clinical training support | 6 | 24 | 19 | 25 | 13 | 87 |
| Initial housing support | 25 | 18 | 6 | 14 | 9 | 72 |
| Professional development support | 11 | 11 | 20 | 9 | 17 | 68 |
| Line management support | 9 | 9 | 10 | 21 | 18 | 67 |
| Relocation allowance | 12 | 17 | 11 | 5 | 16 | 61 |
| Prior relationships (already knew people) | 10 | 10 | 9 | 9 | 13 | 51 |
| Easy adaptation to technology/ equipment | 7 | 8 | 13 | 12 | 11 | 51 |
| Effective communication | 3 | 5 | 12 | 16 | 12 | 48 |
| Ease of recruitment and on boarding process | 9 | 7 | 8 | 12 | 6 | 42 |
| Socialisation and adapting to the UK culture | 2 | 10 | 1 | 9 | 7 | 29 |
| Access to information | – | 6 | 6 | 6 | 10 | 28 |
| Internationally trained staff support group | 2 | 1 | 7 | 4 | 10 | 24 |

*'The most important thing is the accommodation. Some places like my trust [employer] don't give relocation allowance which is meant to help the person adapt properly to the environment. If that could be made available it will allow easy entrance and adjustment to the environment other than struggling with to fit into the economy.'* (ID140)

With regards to equity of opportunity for career development (53.8% agreement) and support to achieve career aspirations (66.7% agreement), these did not rate as positively with respondents as the experiences of transition to work (Fig 4). However, 85.7% of respondents felt motivated to contribute to service delivery and patient care in their department.

**Additional qualitative comments.** Respondents were given the opportunity to add any further comments regarding their experience as an IEDR working in the UK at the end of the survey. Some raised challenges relating to discrimination, singling out or hostility and described about the impact it had on them.

*'I also find that [the] UK is generally more hostile to immigrants. The media have certainly not helped this situation. [It] was worst around brexit time.'* (ID61)

*'I wasn't welcomed, and I have tried to advance in my practice which is very difficult as a result of racial discrimination….. It brought fear and anxiety into me and I was scared of going to work. The worst experience was during pregnancy, it was hell, but I encouraged myself. I sorted help from occupational health and my midwife because I almost committed suicide. I was pleased to be supported by my midwife and occupational health mental nurse'* (ID121)

*'The fact that my staff ID [identity badge] has my listed profession as 'International Radiographer' is evidence that migrant workers who pack up their lives and leave all their possessions behind to start over yet be treated as lesser than the other radiographers. Don't see British new starters ID having 'Local radiographer' so why should migrants ID have'international''* (ID213)

*'Many international trained recruits from Nigeria suffer subtle discrimination in the hands of their managers and colleagues on a daily basis.'* (ID80)

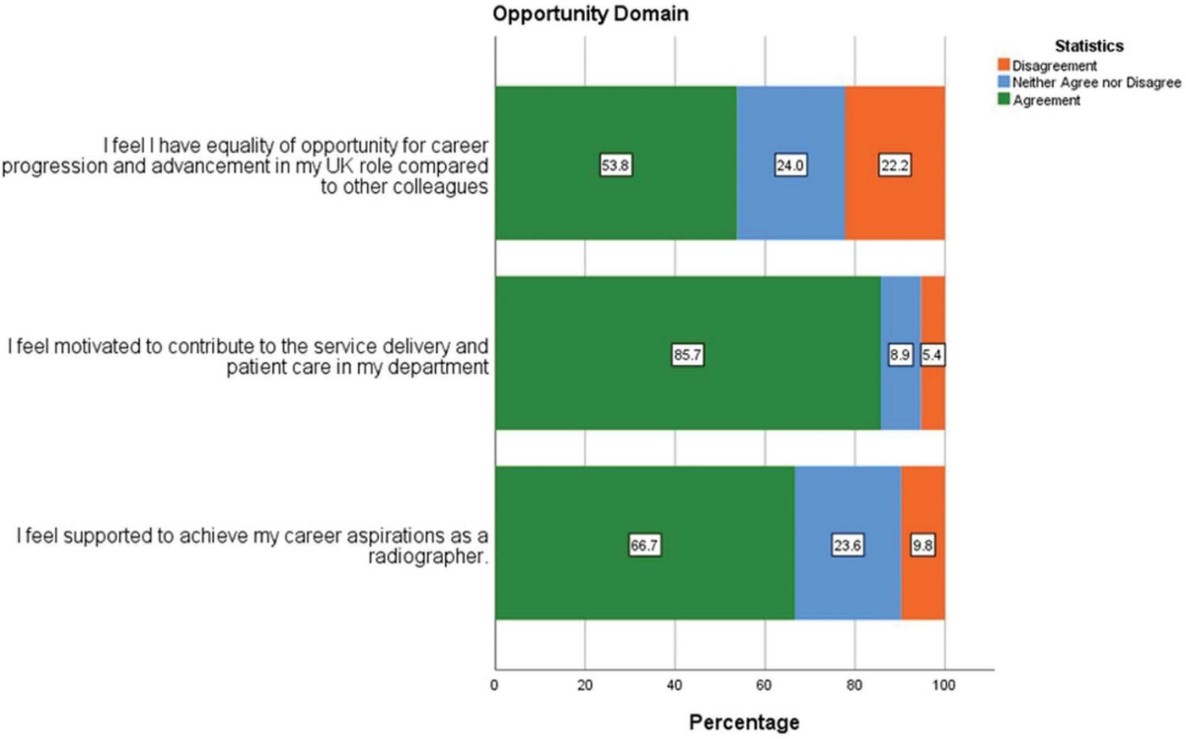

**Fig 4. Responses related to professional opportunity and motivation at work.**

Other respondents felt that more work needed to be done to better induct and support the transition of IEDR into the workplace.

*'I like to put forward few suggestions… a fixed short term mandatory training program which includes the introduction of different protocols of the trust and techniques. [A] few sections [sessions] with reporting radiographers to learn about assessing the image characteristics and to know about what reporters are expecting regarding diagnostic quality of [the] X-Ray. More shadowing in all different shifts… A lecture about commonly used medical terminologies in radiology.'* (ID47)

Discrepancies in the skills and training that IEDR were able to use in their roles in their home country compared to the UK was raised, along with skills not being recognised.

*'I've found that here, compared to home, having the 'right' paperwork is more important than experience and this I think it is a downfall. People who would be better in senior positions actually lose out to those who have the right papers'.* (ID61)

*'I have changed numerous jobs within the NHS, some were better than others. [The] NHS varies a lot but I have been supported in my roles and there is much more career development here although the pay does not reflect this in the UK compared to the same work performed in Australia.'* (ID2)

Some respondents reported the unexpected financial challenges of living in UK.

   

*'UK salary is higher but so is the cost of living'* (ID221)

*'Cost of living is very high in [the] UK. Difficult to find houses. Rent is very high. Very high tax. Difficult to survive here….. Wrong decision to move here.'* (ID 35)

Additionally, some respondents added comments about the personal impact and adjustment to moving to another country.

*'I would like to encourage you to go deep into the emotional balance and relocation changes every radiographer faces when moving to a new country cause [because] it is difficult for every one of them.'* (ID57)

*'It takes time to get adjusted to the UK life and weather, but I am quite happy now that I moved to here.'* (ID34)

## Discussion

There is paucity of research regarding the experiences of diagnostic radiographers who have migrated to the UK to join the healthcare workforce, so these findings offer valuable insights into their transition, integration and intentions to remain. Whilst the ethical issues of migration and recruitment of oversees healthcare professionals and the negative impacts of brain drain are acknowledged [7,8], these were not explored within the survey.

### Transitional experiences

Many respondents reported not knowing what to expect when coming to work in the UK, with some relying on information from contacts from their own country who had relocated previously. This echoes experiences of other internationally educated healthcare professionals (IEHCPs) who reported receiving inadequate information and a finding that the reality did not match their expectations [19–21,26]. Successful integration into a new healthcare system and role is influenced by the level of pre-departure support from the host country's employing organisation [27], thus greater cultural and professional information could better prepare IEDR.

Healthcare professionals coming to work in the UK often experience a difference in the clinical practices [28,29]. Despite the majority IEDR reporting that practice in the UK differed from the practice they had undertaken in their home country, they still felt confident and equipped for the transition based on their training and experience. In addition, the workplace support, mentorship and relationships with colleagues were described positively by the majority of IEDR as aiding adjustment to their new role and the UK. A survey by Alexis [30] of 188 internationally educated nurses working across 15 NHS acute trusts in England also found that respondents felt they received appropriate support in the clinical environment (P < .001). In a study by Leone [31], those good working relationships were shown to make internationally educated nurses feel valued and loyal. This may explain why similarly in this study, those positive work relationships and support meant IEDR were also motivated to contribute to service delivery and care in their departments. Importantly, those who are supported with formal orientation programmes or joined professional organisations experience greater acculturation and commitment to their employers and are more likely to stay [32].

The differences in communication and professional terminology identified in this study are commonly reported as a barrier to transitioning for migrant healthcare professionals [21,22,32,33]. The difficulty radiographers reported in understanding UK healthcare acronyms is echoed in the literature where colloquialisms and medical terminology differences may cause issues or raise concern for patient safety [34]. UK colleagues have also reported that communication difficulties and cultural misunderstandings reduce the efficiency of teams [26]. This need for support around language, communication and UK practice norms has been identified in a study by Golder et al., [12] designed to support integration of diagnostic radiographers trained outside the UK. Language barriers have also been identified as a factor impacting negatively on

 

acculturation [35], and indeed 48 respondents in this study identified effective communication as one of the top 5 factors which supported their integration. The use of an online learning package to raise awareness and give practical support in dealing with challenges and language differences was received positively by IEDR who engaged with it [12] and may indicate such resources are useful in bridging that gap and equipping them for their new role.

## Supporting integration

Family support was rated the most important factor to IEDR's integrating into the UK workforce. Additionally, as in other research, professional associations, networks of people who have already migrated to the UK to work, religious associations or peer support from being part of a larger cohort of internationally educated healthcare professionals recruited to another country, have provided a mechanism for support and integration [21,31,36]. However, a reliance on networks with people from the same nationality can inhibit acculturation [28]. In this study, IEDR of single status had better outcomes in relation to social relationships supporting adaptation to UK life, cultural differences impacting on transition, and loneliness. Al-Hamdan et al., [28] research of nurses originating from one country supports this finding, as those that were single developed social networks and positive interpersonal relationships. For the IEDR who are married this may cause reliance on those relationships and inhibit socialisation and may explain why they are more likely to experience differences in culture affecting adaptation to life and work in the UK. Identifying ways to engage, support and create social networks that can benefit the whole family unit may improve adaptation to UK life, reduce isolation and increase the desire to stay.

Some respondents reported not feeling as though they fit in culturally with the team. Heritage cultural identity salience, the degree to which cultural identity defines self-identity, can influence the acculturation strategies immigrant workers adopt [37]. Where the employing organisations' environment is not supportive of cultural diversity or facilitative of cultural heritage being practised, immigrant employees may be less likely to adopt integration strategies [37]. Therefore, greater acceptance of cultural diversity and the ability for cultural identity to be expressed is needed.

Wider practicalities of relocating to another country were raised, with housing support and relocation allowance ranked in the top 9 factors supporting integration of IEDR. In a study by Bond et al., [18], many organisations recruiting internationally educated nurses and midwives into the UK felt stable accommodation was an important aspect of transition and a tool for integration. Some respondents were keen that the whole upheaval of their lives was acknowledged, and this was not simply just about transitioning into a new job. This was not necessarily couched as negative feelings to living in the UK, as many respondents reported being happy: just the acknowledgement that it was different, and this takes time to assimilate to. O'Brien and Ackroyd [26] identified that considerable practical and emotional support are needed in practice when assimilating overseas nurses into the UK workforce. Calvo et al., research [38] 3 (2024) demonstrates that provision of practical information, such as job opportunities, housing, sourcing goods and facilitating networks, in this case through a WhatsApp group, supports migrant workers to operate in an unfamiliar country. Therefore, the personal challenges [39], emotional burden, as well as practical barriers IEDR will face in relocating to a new country should be anticipated and mechanisms to signpost and support considered.

## Push factors to leave the UK

Prominent pull factors to come to work in the UK reported in the first paper from the survey of this group of IEDR include career opportunities, salary and quality of life [24], mirroring those of other migrant healthcare professionals [19,40,41]. The findings of this study indicate those expectations were not realised by some respondents, as found in other studies [5]. For some, as in other research, despite the wages being higher in the UK, the cost of living countered this and had implications on finances or standard of living [39]. In other studies, nurses coming to work in the UK experienced financial difficulties and found it difficult to live off one wage [28]. For those with spouses and families, visa restrictions may impact on the household income and overall standard of living may not be comparable to their home country, or other countries.

The survey findings identify negative experiences such as bullying, discrimination, and inequity of opportunity for career progression compared to UK colleagues as some of the push factors. Bullying and racism towards internationally educated healthcare professionals is commonly reported [5,30,32,42–44]. As observed in this study, this is more commonly experienced by BAME than white IEHCPs, being ethnic minority groups within the UK [44,45]. Discriminatory and bullying behaviour has also been shown to come from some immigrant healthcare workers of differing ethnic groups who perceive themselves hierarchically superior [46]. The source of bullying in this study indicates some respondents may be limited in tackling the discrimination, as this came from managers, leading them to leave their job or seek support externally. Racism and a lack of respect for diversity within the NHS has been reported elsewhere [44,47–49] and may be more difficult to tackle if it is systemic within an organisation. Bullying and discrimination have shown significant association with higher burnout with consequential links to patient safety due to the impact on performance [45]; and is generally related to poor patient experience [50]. Discrimination experienced by NHS staff has also been shown to result in poorer mental health outcomes [51]. This demonstrates that the impact of discrimination goes beyond inequitable opportunities within the workplace. Multilevel organisational strategies, core leadership with mandated actions and targets that articulates diversity as a high institutional priority is called for in tackling discrimination within the NHS, rather than just mandatory training [52].

The inability to exercise the same clinical skills or have their prior experience recognised in their UK role appears to be a source of frustration. The training of other healthcare professionals educated outside the UK is often questioned, leading to the assignment of more menial tasks compared to their UK counterparts, working below grade, or not utilising the extent of their skills sometimes due to role differences [28,29,32,33,53]. Support to achieve career aspirations and equity for career development opportunities were seen less positively compared to areas of work and personal transition. The Kings Fund [54] identifies that career progression prospects alongside the culture of a team or organisation impacts on retention of staff, and other studies have found career is an important stay factor [40]. Whether a role meets expectations or provides job satisfaction could therefore influence IEDR attrition, thus managers should be cognisant of career planning beyond initial recruitment and induction.

## Limitations

The survey response rate was lower than anticipated, although comparative with response rates from larger professional healthcare groups [55]. Additionally, the self-reported nature of the survey may introduce social desirability bias, though the consistency of experiences reported by IEDR in this survey align well with other studies exploring other internationally educated healthcare professional groups working in the UK.

Although face validity was assessed through researcher and stakeholder review we did not seek to assess consistency or reliability of the survey tool as a whole, although this would be beneficial to support future research in this area. Some respondents did not use consecutive numbers in ranking factors important to their integration, thus the survey instructions may not have had the clarity required despite piloting. In order to avoid making assumptions about the rank number used, these responses were not included in the data.

## Conclusion

With NHS recruitment strategies actively seeking greater numbers of internationally educated diagnostic radiographers to increase workforce numbers, it is important that integration and retention of those staff is given as much, if not greater, priority than the recruitment process. Encouragingly, many IEDR reported positively on employers and colleagues supporting them to adjust and address the anxieties of a new workplace. Preparation for expectations of working in the UK, communication and professional language, effective onboarding processes and appropriate mentorship are some of the ways the NHS and private healthcare providers can better support IEDR's to transition into their new roles. However, with over 40% of IEDR in this study intending to leave the UK or uncertain if they will stay, the challenges, inequity of opportunity to progress careers, and negative experiences of discrimination and bullying are impacting on retention of this workforce. To reduce the push factors of working in

the UK and promote retention, strategies therefore need to look beyond preceptorship to ensure career planning and continuing development are embedded to support fulfilment of aspirations. The findings indicate that cultural and societal acclimatisation is challenging, and the enormity of relocating to another country requires holistic support for individuals, not just the additional support identified in the workplace. Therefore, recruitment and onboarding processes should consider wider aspects to support, with partnerships and signposting to other relevant agencies or groups. Considering support and integration for the wider family unit as well as the IEDR may prevent those IEDR who are married encountering fewer positive experiences to integration. Most importantly, bullying and discrimination policy needs to be authentically embedded and institutional culture tackled to ensure the experiences of IEDR are positive, supportive and welcoming. Bullying and discrimination not only impacts on the individuals who experience it but has repercussions for healthcare delivery and capacity if the expertise is lost.

### Implications for practice

Facilitating good workplace relationships, mentoring and clinical training support can support integration into the UK healthcare workforce. Additional information to support preceptorship can better prepare IEDR for what to expect when coming to live and work in the UK to aid transition. IEDR's have demonstrated a motivation to work in the UK, and the capability and desire to further develop their careers. Capitalising on this with supportive career planning frameworks would maximise the potential of staff, give job satisfaction and enhance service provision. Further work on cultural competence is needed within the UK workforce to prevent bullying behaviours. Additionally facilitating acculturation to reduce isolation and promote retention of internationally recruited staff in the longer term is needed. This will provide a more stable and harmonious workforce and an environment in which IEDR's can thrive, from which service delivery and patient care can benefit. Further research exploring the experiences of employers supporting the integration of IEDR into the UK radiography workforce would be valuable and provide opportunity for triangulation of findings.

### Supporting information

**S1 Table.  S1 Table.pdf Survey questions reported on in this article.**
(PDF)

### Author contributions

**Conceptualization:** Elaine Wilkinson, Edozie Iweka, Beverley Snaith, David Omiyi.

**Data curation:** Elaine Wilkinson, Edozie Iweka, Beverley Snaith, David Omiyi.

**Formal analysis:** Elaine Wilkinson, Edozie Iweka, Beverley Snaith, David Omiyi.

**Investigation:** Elaine Wilkinson, Edozie Iweka, Beverley Snaith, David Omiyi.

**Methodology:** Elaine Wilkinson, Edozie Iweka, Beverley Snaith, David Omiyi.

**Writing – original draft:** Elaine Wilkinson.

**Writing – review & editing:** Edozie Iweka, Beverley Snaith, David Omiyi.

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
