## [Decision Letter · Decision Letter 0]

26 Mar 2025

Dear Dr. Wilkinson,

Thank you for submitting your manuscript to PLOS ONE. After careful consideration, we feel that it has merit but does not fully meet PLOS ONE’s publication criteria as it currently stands. Therefore, we invite you to submit a revised version of the manuscript that addresses the points raised during the review process.

We look forward to receiving your revised manuscript.

Kind regards,

Jordan Llego, PhD ELM, D. Hon. Ex., PhDN, RN

Academic Editor

PLOS ONE

2. In the online submission form, you indicated that [The data underlying the results presented in the study are available from the corresponding author].

Additional Editor Comments:

The paper presents original, timely, and relevant research on an underexplored topic with significant policy implications. However, several methodological clarifications must be addressed, particularly about qualitative analysis, survey design, and the rationale for specific demographic data. Incorporating deeper contextual and comparative insights would also benefit the discussion.

To strengthen the research, it is important to clarify the statistical analysis by stating whether any missing data were handled using specific techniques or ignored on a case-by-case basis.

Additionally, consistency in terminology is crucial; terms such as “IEDR,” “internationally educated diagnostic radiographers,” and “international recruits” should be used uniformly throughout the paper. Furthermore, the conclusions could be expanded to better connect the findings with policy implications, especially concerning NHS workforce planning, onboarding processes, and inclusion efforts.

Lastly, a thorough check on the formatting of references is necessary to ensure consistency and completeness across all entries in the reference list.

Reviewers' comments:

Reviewer's Responses to Questions

**Comments to the Author**

1. Is the manuscript technically sound, and do the data support the conclusions?

Reviewer #1: Yes

2. Has the statistical analysis been performed appropriately and rigorously?

Reviewer #1: Yes

3. Have the authors made all data underlying the findings in their manuscript fully available?

Reviewer #1: No

4. Is the manuscript presented in an intelligible fashion and written in standard English?

Reviewer #1: Yes

Reviewer #1: Title: Integration experiences of internationally educated diagnostic radiographers working in the UK

Thank you for the opportunity to review this interesting paper.

Abstract:

Please add which year the data was collected.

Please add an aim or objective.

Introduction

This sentence is unclear, please consider rephrasing (line 72): however little exists regarding the experiences of internationally educated diagnostic radiographers.

Please consider explaining in depth why this focus is interesting/relevant?

Method

Include which year the institutional ethical approval was obtained and by whom.

Please consider including the survey as appendix

How many piloted the survey-and did that lead to any changes?

Was I possible to skip items?

Why collect data on ethnicity – this is prohibited in EU (I know it is not prohibited in UK, but you are not data collecting in the UK). What is the relevance?

I would think educational backgrounds were more important than ethnicity?

There seems to be many qualitative responses – did you perform an qualitative analysis as well as the quantitative calculations? Please explain.

Discussion

Over all an nice discussion with relevant point for the UK. However, what about comparison with US or the Arab countries (Saudi, Abu Dhabi etc.)? Both countries recruit foreigners to healthcare.

What about moral obligation to the countries recruiting foreigners – any through about brain drain?

Limitations – was the validation of the survey limited?

Is this a limitation – consider to delete? further research exploring the experiences of employers supporting the integration of IEDR into the 389 UK radiography workforce would be valuable and provide opportunity for triangulation of findings.

**Do you want your identity to be public for this peer review?** For information about this choice, including consent withdrawal, please see our Privacy Policy

Reviewer #1: No

---

## [Author Response · Author response to Decision Letter 1]

6 May 2025

Reviewer comment Response from authors

Abstract - add year data collected 2023 has been added

Abstract - add an aim or objective Aim added to abstract

Introduction line 72 –rephase sentence for clarity This has been separated into two sentences and reworded for clarity.

Introduction – explain why this focus is interesting Thank you- additional rationale added to the introduction.

Method – year ethics obtained & by whom Method line 78 wording has been updated to ‘Ethical approval was obtained from the Research Ethics Panel at the University of Bradford on 15th September 2023 by the authors employed there prior to study commencement (EC27952).’

Method - How many piloted the survey-and did that lead to any changes Lines 88-91 have been amended to include the number of individuals involved in the pilot phase and to confirm that the changes related to question wording and survey routing.

Method – Was it possible to skip survey items Results line 22 indicates respondents did not answer all questions but for clarity as you suggest, additional information has been given in the method and a copy of the survey questions has been added as supplementary information (S1 Table).

There seems to be many qualitative responses – did you perform a qualitative analysis as well as the quantitative calculations? Please explain. Additional explanation added to methodology regarding thematic analysis of qualitative free text responses

Why collect data on ethnicity. I would think educational backgrounds were more important than ethnicity? All data collection was undertaken in the UK as the population was internationally educated diagnostic radiographers already working in the UK. Ethnicity as well as country of origin was obtained to explore whether there was a correlation with transition or discrimination as has been reported in other research regarding migrant healthcare workers. Information on education was collected and reported in our already published paper, and we are reporting on different data sets from the survey in this manuscript as explained in method lines 95 & 96.

Consistency in terminology -terms such as “IEDR” Terminology has been changed where appropriate to keep consistency throughout manuscript

To strengthen the research, it is important to clarify the statistical analysis by stating whether any missing data were handled using specific techniques or ignored on a case-by-case basis. Missing data was considered on a case by case basis – this has now been reported in the method section.

Provision of data.

All PLOS journals now require all data underlying the findings described in their manuscript to be freely available to other researchers, either 1. In a public repository, 2. Within the manuscript itself, or 3. Uploaded as supplementary information. Participant privacy in the use of the data was assured in the study information sheet to encourage participation, with information that the data be retained only by the research team. This approach and this wording on the participant information sheet was approved by the Universities Ethics Panel. Specific data is available from the authors on request.

Overall a nice discussion with relevant points for the UK. However, what about comparison with US or the Arab countries (Saudi, Abu Dhabi etc.)? Both countries recruit foreigners to healthcare. Additional references have been added to give a global comparison to UK trends.

What about moral obligation to the countries recruiting foreigners – any thought about brain drain? This is a worthy topic for discussion, however the focus of this paper is integration and experiences for those who have chosen to migrate rather than the issues migration creates. We have published an earlier paper related to a different set of data from the survey which maps where migrant radiographers come from to work in the UK and the drivers. This paper has been referenced in this article.

The conclusions could be expanded to better connect the findings with policy implications This has been added to the conclusion, however, we have not touched on policy regarding international recruitment and its ethics for reasons stated above.

Limitations – was the validation of the survey limited? We have added a sentence to the limitations confirming that we did not seek to validate the survey tool through a formal process but have acknowledged that this should be an aspiration to ensure its applicability going forwards.

Is this a limitation – consider to delete? ‘further research exploring the experiences of employers supporting the integration of IEDR into the 389 UK radiography workforce would be valuable and provide opportunity for triangulation of findings’. This sentence has been removed from limitations and added to the implications for practice section

Check formatting of references- Amended to Vancouver with square brackets as per guidance

---

## [Decision Letter · Decision Letter 1]

13 May 2025

Integration experiences of internationally educated diagnostic radiographers working in the UK

PONE-D-25-03476R1

Dear Dr. Wilkinson,

We’re pleased to inform you that your manuscript has been judged scientifically suitable for publication and will be formally accepted for publication once it meets all outstanding technical requirements.

Kind regards,

Jordan Llego, PhD ELM, D. Hon. Ex., PhDN, RN

Academic Editor

PLOS ONE

Additional Editor Comments (optional):

Thank you for submitting your revised manuscript to PLOS ONE. After thoroughly reviewing your resubmission, including your detailed responses to the reviewers' comments and the revisions you incorporated, I am pleased to inform you that your manuscript has been accepted for publication.

Your study provides significant insights into the experiences of internationally educated diagnostic radiographers (IEDRs) as they integrate into the UK healthcare system. The findings are timely and relevant, especially concerning ongoing workforce shortages and international recruitment strategies. Your rigorous survey methodology, thoughtful discussion of integration challenges, and implications for workforce retention contribute valuable perspectives to health policy and practice.

We appreciate your efforts in addressing the reviewers' feedback. Your enhancements to the methodology, clarification of statistical approaches, and improved terminology alignment have notably strengthened the manuscript. The expanded discussion and conclusion sections connect your findings to broader organizational and cultural integration efforts.

Congratulations to you and your team on this achievement. The manuscript will now proceed to our production department for final formatting and publication. Please ensure all figures, tables, and supplementary files are ready for final typesetting.

Thank you again for choosing PLOS ONE as the platform for your important research. We look forward to this article's contribution to the literature on international healthcare workforce integration.

Reviewers' comments:

Reviewer's Responses to Questions

**Comments to the Author**

Reviewer #1: All comments have been addressed

2. Is the manuscript technically sound, and do the data support the conclusions?

Reviewer #1: Yes

3. Has the statistical analysis been performed appropriately and rigorously?

Reviewer #1: I Don't Know

4. Have the authors made all data underlying the findings in their manuscript fully available?

Reviewer #1: Yes

5. Is the manuscript presented in an intelligible fashion and written in standard English?

Reviewer #1: Yes

Reviewer #1: Thank you for this revision.

The authors have addressed more or less all the points of critic very nicely.

No further comments.

**Do you want your identity to be public for this peer review?** For information about this choice, including consent withdrawal, please see our Privacy Policy

Reviewer #1: No

---

## [Editor Report · Acceptance letter]

PONE-D-25-03476R1

PLOS ONE

Dear Dr. Wilkinson,

I'm pleased to inform you that your manuscript has been deemed suitable for publication in PLOS ONE. Congratulations! Your manuscript is now being handed over to our production team.

Kind regards,

on behalf of

Dr. Jordan Llego

Academic Editor

PLOS ONE